# DNA Printing Integrated Multiplexer Driver Microelectronic Mechanical System Head (IDMH) and Microfluidic Flow Estimation

**DOI:** 10.3390/mi12010025

**Published:** 2020-12-29

**Authors:** Jian-Chiun Liou, Chih-Wei Peng, Philippe Basset, Zhen-Xi Chen

**Affiliations:** 1School of Biomedical Engineering, Taipei Medical University, Taipei 11031, Taiwan; cwpeng@tmu.edu.tw (C.-W.P.); hoppyshi@gmail.com (Z.-X.C.); 2ESYCOM, Université Gustave Eiffel, CNRS, CNAM, ESIEE Paris, F-77454 Marne-la-Vallée, France; philippe.basset@esiee.fr

**Keywords:** DNA printing, flow estimation, MEMS

## Abstract

The system designed in this study involves a three-dimensional (3D) microelectronic mechanical system chip structure using DNA printing technology. We employed diverse diameters and cavity thickness for the heater. DNA beads were placed in this rapid array, and the spray flow rate was assessed. Because DNA cannot be obtained easily, rapidly deploying DNA while estimating the total amount of DNA being sprayed is imperative. DNA printings were collected in a multiplexer driver microelectronic mechanical system head, and microflow estimation was conducted. Flow-3D was used to simulate the internal flow field and flow distribution of the 3D spray room. The simulation was used to calculate the time and pressure required to generate heat bubbles as well as the corresponding mean outlet speed of the fluid. The “outlet speed status” function in Flow-3D was used as a power source for simulating the ejection of fluid by the chip nozzle. The actual chip generation process was measured, and the starting voltage curve was analyzed. Finally, experiments on flow rate were conducted, and the results were discussed. The density of the injection nozzle was 50, the size of the heater was 105 μm × 105 μm, and the size of the injection nozzle hole was 80 μm. The maximum flow rate was limited to approximately 3.5 cc. The maximum flow rate per minute required a power between 3.5 W and 4.5 W. The number of injection nozzles was multiplied by 100. On chips with enlarged injection nozzle density, experiments were conducted under a fixed driving voltage of 25 V. The flow curve obtained from various pulse widths and operating frequencies was observed. The operating frequency was 2 KHz, and the pulse width was 4 μs. At a pulse width of 5 μs and within the power range of 4.3–5.7 W, the monomer was injected at a flow rate of 5.5 cc/min. The results of this study may be applied to estimate the flow rate and the total amount of the ejection liquid of a DNA liquid.

## 1. Introduction

Inkjet print head technology is very important, and the huge development of inkjet technology mainly started with the development of the principle of inkjet print head technology. Contains large-scale droplet generator for studies of inkjet printing [1,2,3,4,5,6,7,8]. The continuous inkjet system has the advantages of high frequency response and high-speed printing. However, the structure of the inkjet print head of this method is more complicated, and it requires a pressurizing device, a charging electrode, and a deflection electric field, which is difficult to mass produce. The inkjet print head of the on-demand inkjet system is simple in structure, easy to realize the multi-nozzle of the inkjet head, easy to digitize and color, and the image quality is relatively fine, but the general ink droplet ejection speed is low [9,10,11].

The total number of nozzles of the hot bubble inkjet head can reach hundreds or even thousands. The nozzles are quite fine, which can produce rich harmony colors and smoother mesh tone. The ink cartridge and the nozzle form an integrated structure, and the inkjet head is updated at the same time when the ink cartridge is replaced, so that there is no need to worry about the nozzle clogging, but it causes waste of consumables and relatively high cost. The on-demand inkjet technology ejects ink droplets only in the graphics and text parts that need to be ejected, and no ink droplets are ejected in the blank areas. This jetting method does not need to charge the ink droplets, and also does not need to charge electrodes and deflection electric fields. The nozzle structure is simple, and it is easy to realize the multi-nozzle of the nozzle, and the output quality is more refined; through pulse control, digitization is easy. However, the ejection speed of ink droplets is generally low. There are three common types: thermal bubble inkjet, piezoelectric inkjet, and electrostatic inkjet. Of course, there are other types.

The realization principle of piezoelectric inkjet technology is: placing many small piezoelectric ceramics near the nozzle of the print head, the piezoelectric crystal will deform under the action of an electric field. Extruded from the ink cavity, ejected from the nozzle, the pattern data signal controls the deformation of the piezoelectric crystal, and then controls the amount of ink ejection. Drop-on-demand hybrid printing using piezoelectric MEMS printhead [12]. The realization principle of thermal bubble inkjet technology is: under the action of heating pulse (recording signal), the temperature of the heating element on the nozzle rises, causing the ink solvent nearby to vaporize to generate a large number of nucleated small bubbles. The volume of the inner bubble continues to increase. When it reaches a certain level, the pressure generated will cause the ink to be ejected from the nozzle, and finally reach the surface of the substrate, reproducing the pattern information [13,14,15,16,17,18]. 

The meanings of “3D product printing” and “incremental rapid manufacturing” have evolved and refer to all incremental product manufacturing technologies. Although this has a different meaning from the previous production, it still reflects the common feature of the 3D work production process, which is stacking materials under automatic control [19,20,21,22,23,24].

This development system is thermal bubble jet technology. It is to design different heater diameter and cavity thickness to deploy DNA beads in this rapid array, and to evaluate the spray flow rate. The boost circuit system in the DNA jet chip is the signal source for driving large flow. The purpose is to adjust the amount of DNA solution sprayed and the output power. When the input voltage needs to be converted to a higher output voltage, a boost converter is the only choice. The boost converter achieves the purpose of boost output by charging the voltage through the internal metal oxide semiconductor field effect transistor (MOSFET), and when the MOSFET is turned off, the inductor is discharged through load rectification.

The conversion process between charging and discharging the inductor will reverse the direction of the voltage through the inductor, and then gradually increase the voltage higher than the input operating voltage. The switching duty cycle of the MOSFET definitely determines the boost ratio. The rated current of the MOSFET and the boost ratio of the boost converter determine the upper limit of the load current of the boost converter; the rated voltage of the MOSFET determines the upper limit of its output voltage. Some boost converters integrate the rectifier and MOSFET to provide synchronous rectification. The integrated MOSFET achieves accurate zero current turn-off, making the boost transformer more efficient. Real-time monitoring of the input power through the maximum power point tracking unit. When the input voltage reaches the maximum input power point, the boost converter starts to work, making the boost converter suitable for DNA printing on glass substrates with maximum power output point. Through the constant on-time generating circuit, the on-time is not affected by temperature and the corner angle of the chip, which enhances the stability of the system.

The technology employed in an inkjet print head is critical. The tremendous advances in inkjet technology primarily began with the development of theories behind inkjet print head technology, including the large-scale droplet ejector used in inkjet printing [1,2,3,4,5,6,7,8]. A continuous inkjet system has the advantages of high-frequency response and high-speed printing. The total number of the nozzles on the inkjet head can reach hundreds or even thousands, and these nozzles are exceedingly intricate. A nozzle can generate rich and harmonious color and smooth mesh tone [9,10,11]. Inkjets can be categorized into three major types: thermal bubble inkjet, piezoelectric inkjet, and electrostatic inkjet. Other types are also in use. Piezoelectric inkjets function as follows: Many small piezoelectric ceramics are placed near the nozzle of the inkjet head. Piezoelectric crystals deform under the electric field. Subsequently, ink is squeezed from the ink cavity and ejected from the nozzle. The data signal of the pattern controls the deformation of the piezoelectric crystals and then controls the amount of ink ejected. Piezoelectric microelectronic mechanical system (MEMS) inkjet heads are used for hybrid printing [12]. Thermal bubble inkjet technology functions as follows: Under a heating pulse (i.e., the recording signal), the temperature of the heating component on the nozzle increases, evaporating the ink solvent nearby, thereby generating a large amount of small nucleated bubbles. The volume of the internal bubbles continuously increases. When it reaches a certain level, the pressure generated leads to the ejection of the ink from the nozzle and the ink reaching the surface of the substrate, thereby presenting patterns and messages [13,14,15,16,17,18]. 

The evolution of three-dimensional (3D) product printing and rapid prototype techniques involves the production technology of all rapid prototypes. Although the rapid prototype technique differs from conventional production, it shares some characteristics of the production process of 3D product printing. Specifically, it stacks materials under automatic control [19,20,21,22,23,24]. 

The system developed in this study employed thermal bubble ejection technology. Different diameters for the heater and different cavity thickness were employed to deploy DNA beads in this rapid array. Subsequently, the spray flow rate was evaluated. The boosted circuit system in a DNA jet chip is the signal source for driving large flow. The goal is to adjust the amount of DNA liquid being sprayed and the output power. When the input voltage must be modified to a higher output voltage, a step-up converter is the only option. A step-up converter charges the internal metal oxide semiconductor field effect transistor (MOSFET) to enable an increased output voltage. When the MOSFET is turned off, inductance is discharged through load rectification. The process of changing an inductor between charging and discharging alters the direction of the voltage through the inductor. The voltage gradually increases to the point that it exceeds the input operating voltage. The duty cycle of the MOSFET’s switch determines the boost ratio. The rated current and the boost ratio of the step-up converter of the MOSFET determines the upper limit of the load current of the step-up converter. The rated current of the MOSFET determines the upper limit of its output voltage. Some step-up converters integrate rectifiers and MOSFET to provide synchronous rectification. An integrated MOSFET can realize precise zero current shutdown, thereby making the set-up converter more efficient. The maximum power point tracking unit was used to monitor input power in real time. When the input voltage reaches the maximum input power point, the step-up converter starts operating. The step-up converter is used on the glass substrate with a maximum power output point for DNA printing.

## 2. MEMS Chip Design for Bubble Jet 

This study designed five types of liquid ejection chamber structures by manipulating certain parameters, such as heater size, heater number, and loop resistance. Table 1 lists the measurement results. That system analyzed the loop resistance of different heaters. Because it passes through each single circuit series using two sets of heaters to complete the 100-heater design, when designing 100 heaters, the total loop resistance must be one heater larger than the total loop resistance of 50 heaters. In this study, during the process of bubble ejection from the MEMS chip, the sheet resistance of the resistance layer was 29 Ω/m^2^. Therefore, the total loop resistance of Model A was the largest. It was twice that of the normal size models (Models B1, C, D, and E). The total loop resistance of Models B1, C, D, and E was approximately 29 Ω/m^2^. According to Table 1, its error range was within the allowed design value. Thus, the single chips of each type designed in this study had the same production procedure results, and they were used for subsequent flow rate measurement.

Once the power in the DNA-sprayed chip was confirmed to be normal, the characteristics of the growth of the heater bubbles were tested and verified. The film thickness and film quality of the DNA-sprayed chip affect the operating conditions and spraying quality of the heater; therefore, understanding the bubble growth phenomenon and its growth characteristics helps clarify the operating conditions as well as the characteristics of the DNA-sprayed chip in this study. 

The designed system adopted the open liquid supply method to observe the bubble growing conditions. For image observation, the synchronous flash method, employing light-emitting diodes (LED, Nichia NSPW500GS-K1, 3.1 V White LED 5 mm), was used to generate a synchronous delayed light source. The system also used a charged-coupled device (CCD, Flir Grasshopper3 GigE GS3-PGE-50S5C-C) to capture images. Figure 1 demonstrates the process of a bubble from nucleation, growth, bubble generation, to dissipation. The system confirmed the growth and dissipation process of bubbles, which can be used for observing starting voltage. Regarding the supply method of liquid in the microchannels, the time when the LED blinks was set as the time required to generate the largest bubble (15 μs). This design prevents incorrect judgments resulting from of an unsuitable blinking time as well as the inability to capture bubble images. 

In the open pool experiment, we set the operating frequency at 5 KHz. By adjusting the pulse width, we measured the starting voltage of various heaters. Table 2 presents the testing results.

Application-specific integrated circuit (ASIC) chip was used in the DNA liquid ejection system. This system was developed with the aim to design a specific ASIC chip with different apertures and cavity thicknesses to place DNA beads in this rapid array. 

This study aimed to develop a unique circuit system to manage the digital register for use in designing and producing ASIC chips for DNA liquid ejection. The system structure comprises serial input data and parallel output data. The “register digital circuit” function generates location “A” to search for the activity nozzle. The register serves as a bistable multivibrator model system.

The proposed technology was compared with the regular scan sequence, and the system with the novel technology only required approximately half the time of the conventional sequence. The serial input data and parallel output data are connected with a shift register (Figure 2). The input is a serial data bit, which is fed to the parallel-bit data output and connected to the boost circuit. Once the information has been input, the system transmits several serial data bits. The system can obtain the first-level parallel data output. Each trigger is a rising edge trigger model.

## 3. Experiment and Results 

We employed Flow-3D to simulate the internal flow field and flow rate distribution in a 3D spray room. The simulation procedure was used to calculate the time and pressure to generate thermal bubbles. Also, the corresponding mean outlet speed state of the fluid was calculated. The “outlet speed state” function in Flow-3D served as the power source for spraying liquid into a monomer. The actual chip generation process and the starting voltage curve were measured and analyzed. Finally, this study presents the flow rate experiment as well as the results and discussion. 

### 3.1. Flow-3D Simulation

The software simulation was based on the actual DNA liquid cavity calculation to simulate the ejection cavity with single-channel supplying liquid. The geometric shape of the ejection cavity is illustrated in Figure 3. 

The diameter of the ejection outlet was 60 μm. The thickness of the ejection orifice (volcano) was 50 μm. The process thickness of the dry film layer was 60 μm. The bottom design of the ejection cavity was 120 μm × 120 μm. The area of the heater was 105 µm × 105 µm. Under the operating frequency of 5 KHz, the 3D side view and the X-Z two-dimensional (2D) section view were analyzed. Simulation spraying was conducted for 10, 20, 30, 40, and 200 µs. This simulation calculated the area of the top of the thermal resistance. Initially, the mean droplet tip speed was estimated at 10 m/s, and the length of the tail of a droplet was approximately 300 μm. A 200-μm X-Z section view revealed the simulation results. The liquid clearly reached the fluid outlet. The liquid was in a stable state, indicating that the liquid had filled up the cavity as expected. The cavity had a uniquely designed shape. The simulation results revealed the behavior of the geometric shape of the 3D ejection cavity. The simulation results under a 5-kHz operating frequency revealed no shortage of DNA liquid. Therefore, the tail of the DNA droplet had the lowest quality and was easily affected by the asymmetry of the flow field force, resulting in an off-track flying trajectory.

Figure 3 reveals an obvious deflection in the droplet tail of the supplied liquid at 40 μs. As the flight time increased, the horizontal displacement deflection became more substantial. In inkjet printers or other industrial-use inkjet printing processed, the asymmetrical geometric shape of the relative ejection direction (Z-axis direction) must be redesigned to restore the ejection cavity. The direction-related problem did not affect the realization of the planned goals. Figure 4 reveals that, with an orifice diameter of 60 μm and a thickness of 50 μm, the 2D X-Y cross section was higher. The 2D cross-sections before and after the fluid ejection cavity was ejected under instant flow field could be clearly observed. In addition to the orifice, liquid droplets were ejected. The fluid ejection cavity pushes the liquid from the single channel. The results revealed that at 20 μs, the thermal air bubbles enter a state of dissemination. During that stage, fluid starts to refill, and the speed vector passes inside the cavity and points at the outlet channel. During the 30-μs and 40-μs stage, because liquid has been ejected from the orifice, a certain amount of liquid enters the nozzle, and the cavity is not completely filled with liquid. Under this circumstance, external air passes the orifice and enters the cavity while waiting for the fluid to pass the refilled object, and then wait to flow out through the outlet. If air is not completely ejected, it may block the bubbles. Bubbles may also occupy the injection cavity; hence, during the next injection, an adequate volume of liquid cannot be ejected. In addition, because fluid attenuates the tremendous amount of heat generated from the heater, it has a cooling effect. Therefore, the injection cavity inside the chip may accumulate heat. This harmful cycle reduces the ejection performance, finally resulting in the inability to eject liquid droplets; the excessive internal heat also damages the chip. Microfluid channels are not ideal for the injection cavity design. This may be because the inside retains bubbles, resulting in reduced performance. If the heater does not immediately cease operation, the excessive temperature can damage the chip. A major part of the overall design parameters is focused on addressing this concern.

Figure 5 depicts the calculation results of the 2D X-Z cross section. At 100 μs and 200 μs, the fluid injection orifice did not completely fill the chamber. This may be because the size of the single-channel injection cavity was unsuitable for the highest operating frequency of 10 KHz. Thus, subsequent calculation simulations employed 5 KHz as the reference operating frequency. The calculation simulation results were calculated according to the operating frequency of the impact. Figure 6 illustrates the injection cavity height as 60 μm and 30 μm and reveals the 2D X-Y cross section. At 100 μs and 200 μs, the fluid injection orifice did not completely fill the chamber. In those stages, the fluid was still filling the chamber, and the flow field was not yet stable.

### 3.2. Starting Voltage Curve

The DNA printing integrated multiplexer driver MEMS head (IDMH) follows the 0.35-um 2P2M processing model standard complementary metal–oxide–semiconductor (CMOS) design rules to establish the model (Figure 7). This study involved high-pressure processing. For systems based on the 0.35-um DPDM 12 V/12 V or 12 V/5 V mixed model processing model design and allocating high voltage circuit, this study design may serve as a basic allocation guide. This study investigated the differences from the original 0.35-um CMOS mixed-mode (DPDM, 5 V) design principles. That is, during this process, we specified NDD, PDD, HV_OX rules, and high voltage equipment rules. Basically, the low voltage (LV) part adopts the 5-V logic, and its allocation rules are the same as those for the Taiwan Semiconductor Manufacturing Company common 0.6-um DPDM (including the PO1/PO2 capacitor options) ASIC design rules. 

The measurement of the starting voltage curve was conducted on the ASIC on the actual manufactured MEMS chip. Figure 8 and Figure 9 reveal the starting voltage and energy of the Type A chip, respectively. That chip was designed using dual heaters in series. The figures reveal that the starting voltage of the chip decreased as the pulse width increased. From the energy perspective, an increase in pulse width increased the required energy. 

Figure 8 depicts the starting voltage graphs of Type B and Type E chips. The nozzle density of these two chips was 50, and their resistance was 35 Ω. As the pulse width increased, the starting voltage of Type B and Type E chip decreased. From the energy perspective, an increase in pulse width increased the required energy (Figure 9). The Type E design required more energy than did Type B. Although the two designs have the same circuit resistance and the same heater resistance (with an L/W value of 1), the area of Type E is greater than that of Type B. Therefore, regarding the power density of the work performed by the heater, when the same driving pulse voltage was input into these two heater designs, the power density of the work performed by the Type E heater was higher than that of Type B. Type B was designed very high; thus, the starting voltage curve of Type E is higher than that of Type B. Figure 8 illustrates the starting voltage of chip Type C and Type D. Both chips have a long heater design, and they have the same heater area. The L/W value of these two models was 0.75 and 1.33, respectively. The figure indicates that as the pulse width increased, the starting voltage of Type C and Type D declined. The starting voltage of Type D was higher than that of Type C. Concerning energy, an increase in pulse width increased the required energy (Figure 9). The energy required for the Type C design is higher than that of Type D. 

### 3.3. Flow Experiment and Results Discussion 

This study endeavored to estimate the amount of DNA being ejected and the amount of the DNA sample for testing. Therefore, we conducted tests on power conditions and DNA flow. 

The spray DNA chip composed of 50 nozzles could be placed rapidly, and the heater area was increased. The maximum flow rate was 3.5 cc/min, and the DNA chip ejection speed was increased. In this study, we increased the density of the unit area nozzles without increasing the size of the chip, and the microchannel design method was used to expand the original two-line nozzles into four-line nozzles. The number of nozzles used on the spray DNA chip was substantially increased from 50 to 100. The increase in nozzle number was for testing maximum flow. The Type A chip had an increased density and 100 nozzles. On a chip with increased nozzle density, we first used chip No. B2-01 to conduct a flow test. We fixed the driving voltage at 25 V for the experiment, and we observed the flow curve under various pulse widths and operating frequencies (Figure 10). 

An experiment was conducted to analyze different pulse widths. When the pulse width was 4 μs, the flow rate of the injected monomer linearly increased at a frequency of 1–2 KHz. The linear slope was 2.22. When the frequency was 2.5 KHz, the flow rate no longer increased linearly. When the operating frequency was increased to 3 KHz, the flow rate began to decrease. When the pulse width was 5 μs, the linearly increased frequency was only 1.5 KHz, and the linear slope was 2.68. When the pulse width was increased to 6 μs, the linearly increased frequency was smaller than 1.5 KHz, and the linear slope was 3. 

We observed the experimental results under an operating frequency of 15 KHz (Figure 11). The volume of the spray dots increased from 330 pl under a pulse width of 4 μs to 500 pl. At 6 μs, the volume increased with the pulse width. The ejection rate of the ejection point’s volume, in contrast to the chamber capacity, increased from 40% to 58%. The experimental results indicated that an increase in energy increased the pressure of bubbles and more liquid was expelled. 

We observed the flow rate changes under various power conditions, with an operating frequency of 2 KHz and a pulse width of 4 and 5 μs. With power ranging from 4.3 W to 5.7 W, the flow rate of the injected monomer was 5.5 cc/min (Figure 12). The experimental parameter was the maximum yield of the DNA liquid. 

This study tested the total flow rate of the ejected DNA chip. First, we injected a monomer in chips in the B1-oo series. Under an operating frequency of 2 KHz, various operating voltages were employed to test the flow rate. The results revealed that the amount of injected monomer was closely related to the provided power from the outside environment (Figure 13). Although the injected monomer had a different pulse width energy, under the same power condition, the energy obtained from injecting the monomer was the same. The bubbles generated on the heater had the same instant pressure. Therefore, the introduced flow rate was similar. The figure revealed that the flow rate increased with the added power. At a power condition of 4.5 W, the system had a maximum flow rate of 3.5 cc/min. When the power exceeded that level, the flow rate decreased. 

We employed the same experimental procedures for C Type-01 and D Type-02 flow rate test chips. The results are presented in Figure 14 and Figure 15, respectively. Under the condition of a fixed operating frequency of 2 KHz, the amount of monomer injected changed according to the external power (Figure 14). The maximum flow rate (3.5 cc/min) of the injected monomer appeared when the power was 4.5 W. Figure 15 revealed the same result: The maximum flow rate (2.5 cc/min) of the injected monomer in D Type-02 was achieved under the power of 4.2 W. At 4-W added power, another D Type-02 chip achieved its maximum flow (3 cc/min). 

In the aforementioned experiments, the chip types were heaters (size: 105 μm × 105 μm) with a nozzle density of 50. The nozzle outlet size was 80 μm for all chip designs. On the basis of the aforementioned results, the maximum flow rate for all B1 series chips was limited to 3.5 cc/min. The required power for generating the maximum flow rate was 3.5–4.5 W. 

In an effort to increase the amount of monomer being injected, we first considered expanding the size of the heater. The unit heater size of chip No. E1-00 was increased from 105 μm × 105 μm to 132 μm × 132 μm. The size of the microfluid flow channel was increased from 120 μm × 120 μm to 150 μm × 150 μm. The injection cavity volume increased by 56%. The number of monomer injection nozzles was maintained at 50 for the experiment. Figure 16 reveals the flow rate test results. Under an operating frequency of 3 KHz and pulse width of 3 μs, the flow rate of the nozzle under 20–25 V of driving voltage range could be observed. The flow rate increased with the voltage. In excess of 25 V, any further voltage increase reduced the flow rate. When the pulse width was changed to 3.5 μs, the flow rate’s increase range was reduced to 20–23 V. When the voltage exceeded 23 V, the flow rate began to decrease. The pulse width was contained to increase to 4 and 5 μs, and the increased voltage range was further reduced. 

We observed flow performance under various pulse widths under 20 V of driving voltage. As the pulse width increased from 3 μs to 3.5 μs, 4 μs, and to 5 μs, the corresponding flow rate increased from 0.75 cc/min to 23 cc/min, 2.7 cc/min, and to 3.1 cc/min. Figure 17 depicts the correlation between power and flow rate. Flow rate was still affected by power. The figure reveals that as the power increased, the flow rate increased from 0.75 cc/min to 3.5 cc/min until the power reached 6.5 W. When power increased further, the flow rate did not increase with the energy, revealing that the maximum flow rate of this design is 3.5 cc/min. 

The summary of the experimental results is obtained from increasing the number of nozzles, increasing the input signal frequency, pulse width, the reduced heater size, and decreasing the input voltage (power) data to obtain the optimal DNA spray. DNA is not easy to obtain, it is estimated the amount of DNA sprayed outcomes from the data (increased number of nozzles, reduced heater size) while quickly deploying DNA according to the input parameters.

Spraying was conducted using a special 3D MEMS structure. DNA spraying technology can be employed to test the relationship between the flow rate of DNA on the glass slide and time. Because DNA cannot be obtained easily, rapidly deploying DNA while calculating the amount of DNA being sprayed is essential. Microarray biochip sample spots can probe clusters composed of DNA, RNA, peptide proteins, antibodies, cells, or human tissues. The sampling material was determined by the biological characteristics to be measured. For example, the DNA chips for testing the single nucleotide A, T, C, or G (single nucleotide polymorphism) are sample points of different single-stranded DNA sequence fragments being grouped; in this case, DNA sequences were used. The paring principle of AT and CG. When the designed single-stranded DNA sample spot can be paired with the sample and emit fluorescence, it indicates that the DNA sequence was changed by a single nucleotide, and the change in the DNA sequence can be used to explore the chromosome genome diversity of species (including humans). Another example is the protein biochip for detecting food-specific allergens. The conventional method is to place dozens to hundreds of different food allergen sampling point on a standard-size 1″ × 3″ glass. A probe design of 200–500 microns and its fluorescence strength can be used to screen and test dozens to hundreds of different food allergens. 

Conventional experimental procedures require a long time to extract and purify DNA (DNA separation), removing polymerase chain reaction inhibitors to obtain DNA. Although DNA extraction is feasible, during the extraction process, precious samples may be lost. Human operation may lead to experimental errors and sample cross contamination. DNA nucleic acid is highly valuable. Through this experiment, we can precisely estimate the total amount of DNA required for thousands of spots on the microarray biochip sprayed with DNA.

## 4. Conclusions

This study provided a DNA printing IDMH and conducted microfluidic flow estimation. Under the designed DNA spray cavity and 20 V of driving voltage, the flow performance of different pulse widths was discovered to increase with the pulse width. The E1 type flow rate test revealed that as the corresponding flow rate increased to 3.1 cc/min, the flow rate was affected by changes in power. As the power increased, the flow rate increased from 0.75 cc/min to 3.5 cc/min, up to a power of 6.5 W. If the power was increased further, the flow rate did not increase with the energy. This reveals that this table design is the largest. The flow rate was 3.5 cc/min. 

In a specifically designed DNA spray room structure with an operating frequency of 2 KHz and pulse widths of 4 μs and 5 μs, we observed changes in flow rate under various power conditions. Within the power range of 4.3–5.87 W, the flow rate of the injected monomer was 5.5 cc/min. This did not change as the power increased. DNA is precious and cannot be obtained easily. Through this experiment, we can precisely estimate the total amount of DNA required for thousands of spots on the microarray biochip sprayed with DNA

## Figures and Tables

**Figure 1 micromachines-12-00025-f001:**
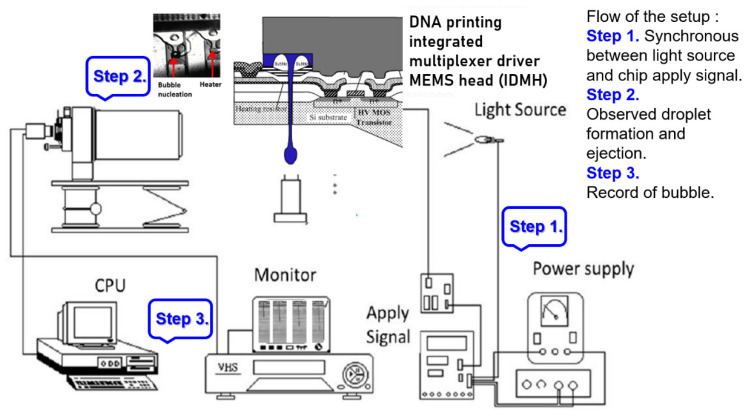
The system uses CCD to capture images. The process of bubble from nucleation, growth, bubble generation to dissipation (Flow of the setup: Step 1. Synchronous input power signal (for light source) and chip drive pulse (Apply signal), Step 2. Observe droplet formation and ejection trajectory through a microscope Observe droplet formation and ejection trajectory through a microscope, Step 3. Process record of bubble from nucleation, growth, bubble generation to dissipation).

**Figure 2 micromachines-12-00025-f002:**
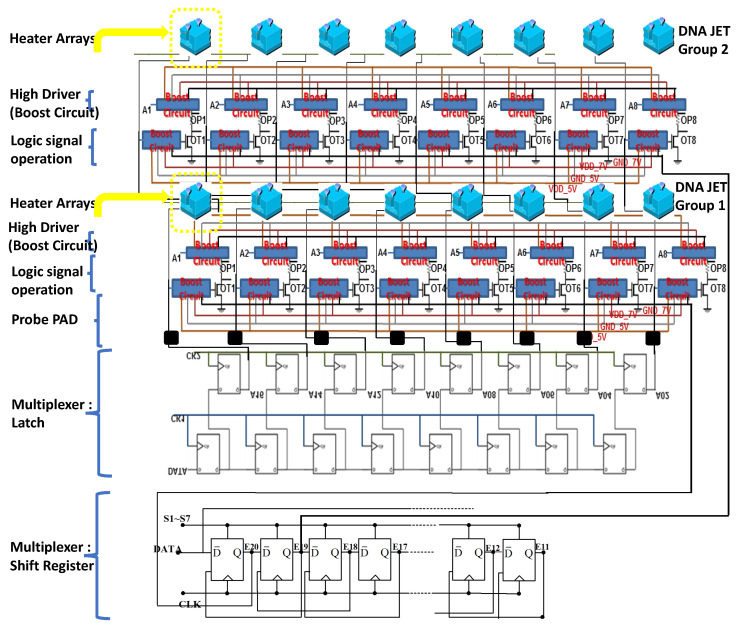
Serial input parallel output shift registers forms of connection.

**Figure 3 micromachines-12-00025-f003:**
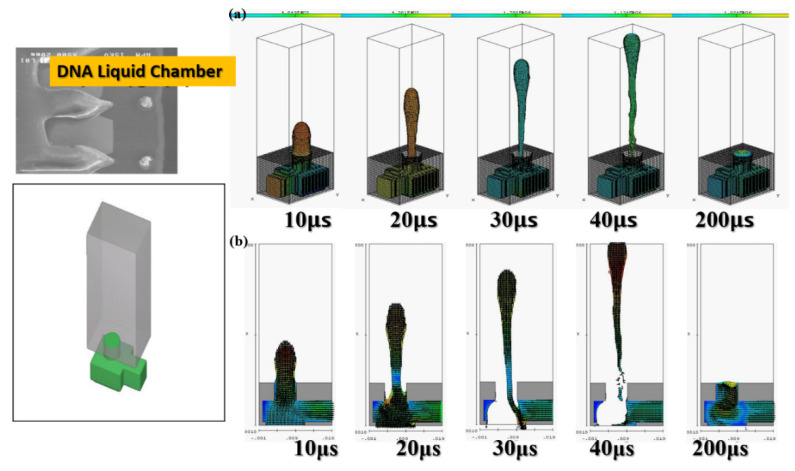
The geometry of the jet cavity. (**a**) The actual DNA liquid chamber, (**b**) the three-dimensional view of the microfluidic single channel. A single-channel jet cavity with 60 μm diameter and 50 μm thickness, with an operating frequency of 5 KHz, in (**a**) three-dimensional side view (**b**) X-Z two-dimensional cross-sectional view, at 10, 20, 30, 40 and 200 μs injection conditions.

**Figure 4 micromachines-12-00025-f004:**
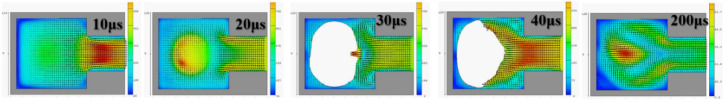
Calculate and simulate the injection of water in a single-channel injection chamber with a nozzle diameter of 60 μm and a thickness of 50 μm, at an operating frequency of 5 KHz, in the X-Y two-dimensional cross-sectional view, at 10, 20, 30, 40 and 200 μs.

**Figure 5 micromachines-12-00025-f005:**
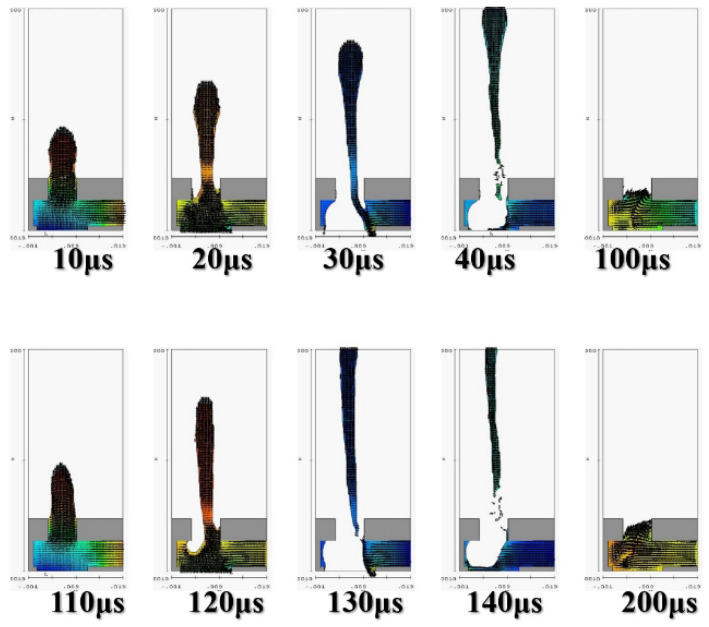
Calculate and simulate water in a single-channel jet cavity with a nozzle diameter of 60 μm and a thickness of 50 μm, with an operating frequency of 10 KHz, in the XZ two-dimensional cross-sectional view, at 10, 20, 30, 40, 100, 110, 120, 130, 140 and 200 μs injection situation.

**Figure 6 micromachines-12-00025-f006:**
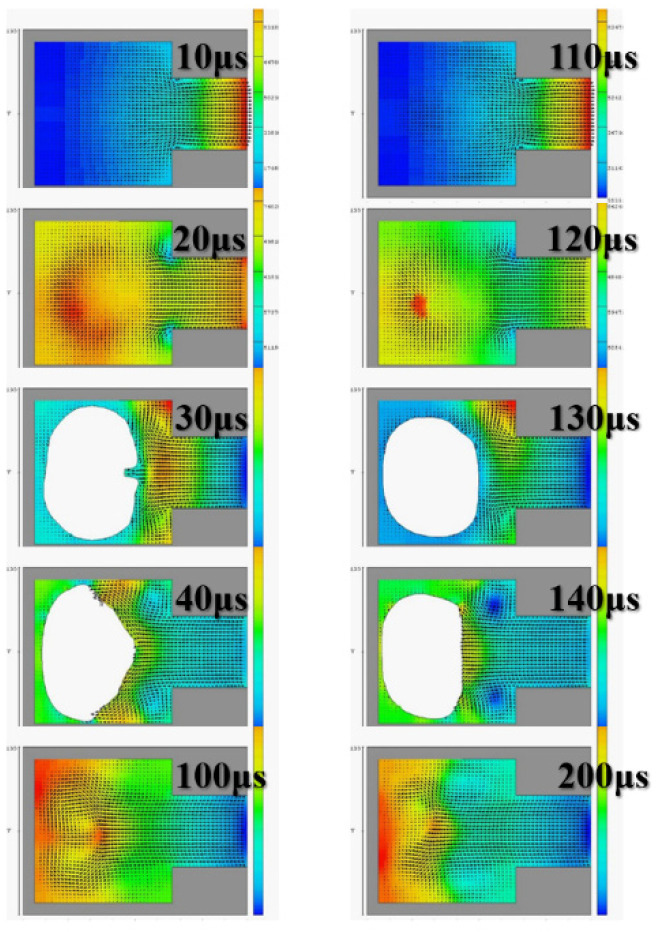
Calculate and simulate water in a single-channel spray chamber with a spray hole diameter of 60 μm and a thickness of 50 μm, with an operating frequency of 10 KHz, in an XY cross-sectional view, at 10, 20, 30, 40, 100, 110, 120, 130, 140 and 200 μs injection situation.

**Figure 7 micromachines-12-00025-f007:**
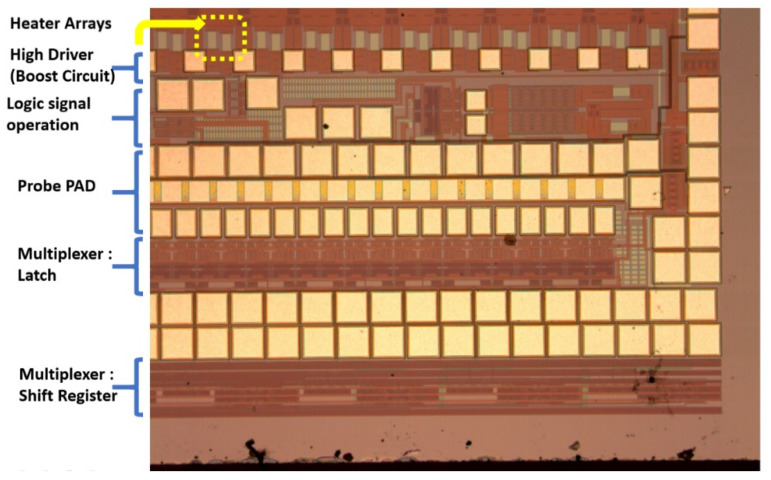
The DNA printing integrated multiplexer driver MEMS head (IDMH).

**Figure 8 micromachines-12-00025-f008:**
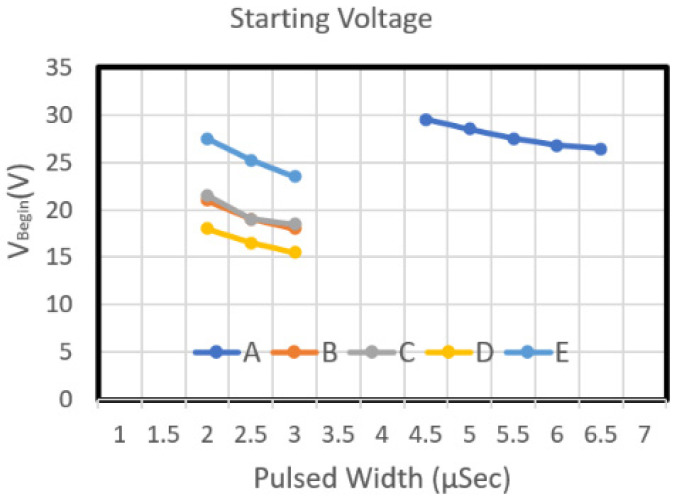
The initial voltage diagrams of chip number A,B,C,D,E type.

**Figure 9 micromachines-12-00025-f009:**
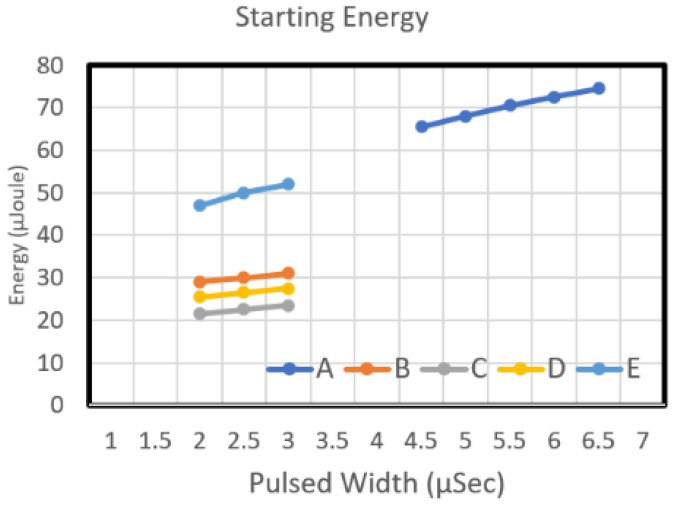
The initial energy diagrams of chip number A,B,C,D,E type.

**Figure 10 micromachines-12-00025-f010:**
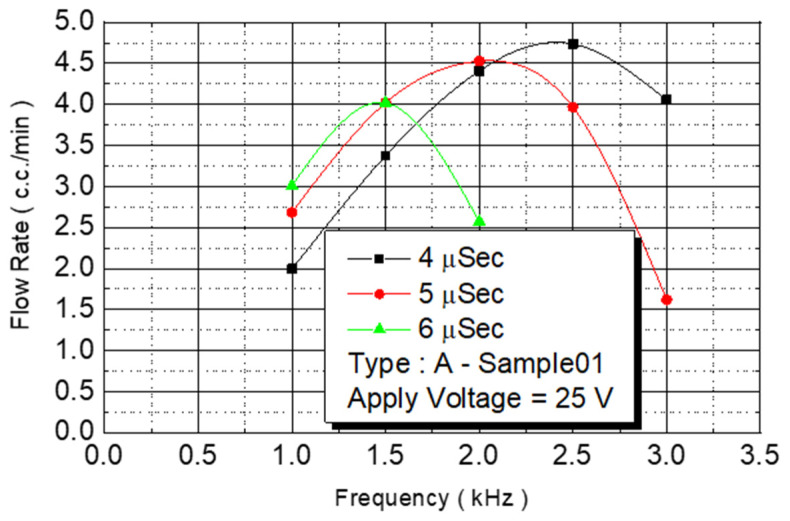
A Type-Sample01 flow test.

**Figure 11 micromachines-12-00025-f011:**
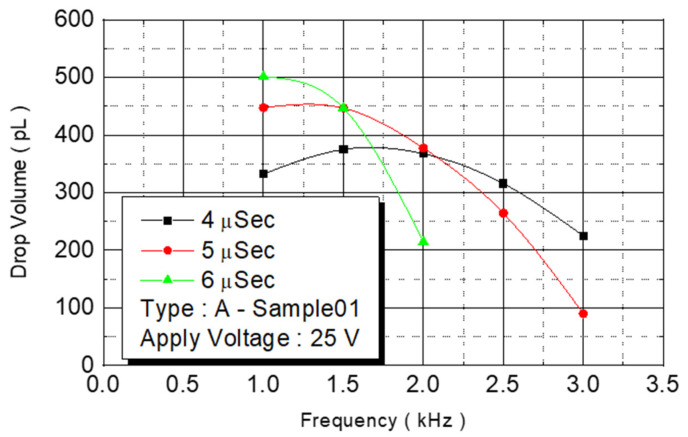
A Type-Sample01 drop volume.

**Figure 12 micromachines-12-00025-f012:**
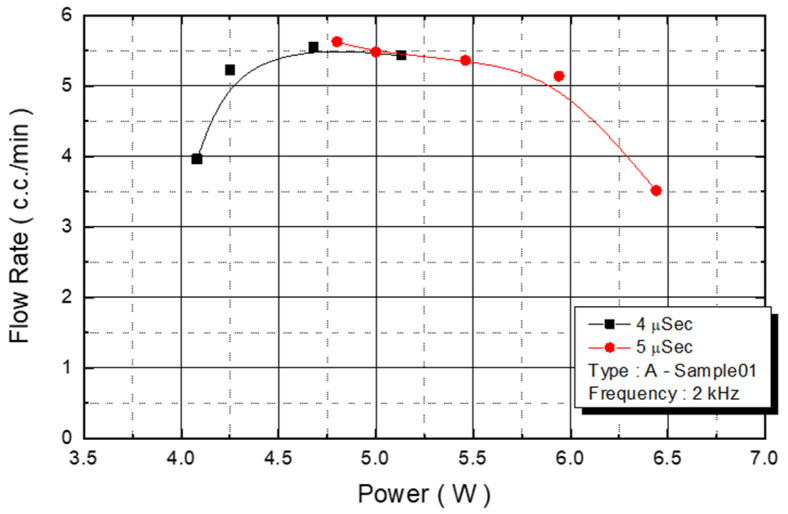
A Type-Sample01 flow rate.

**Figure 13 micromachines-12-00025-f013:**
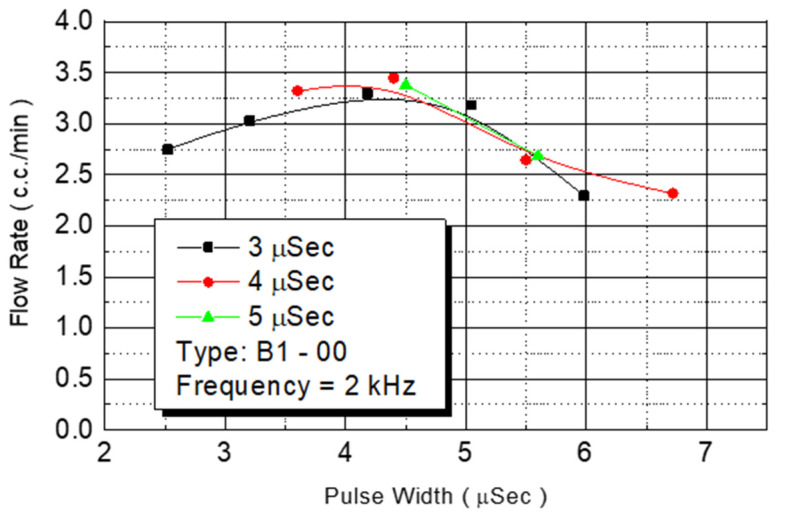
B1-00 flow test.

**Figure 14 micromachines-12-00025-f014:**
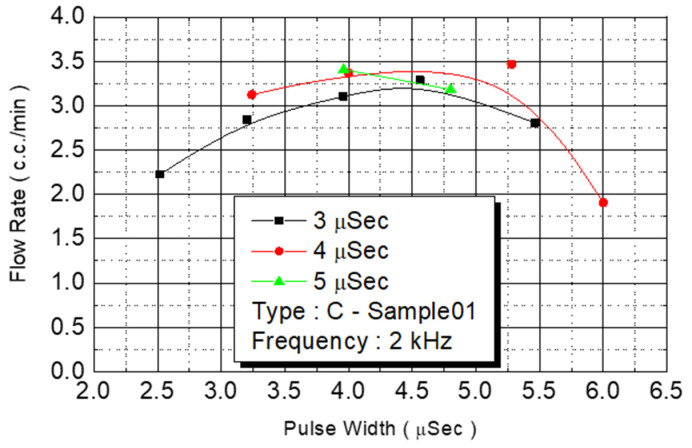
C Type-01 flow test.

**Figure 15 micromachines-12-00025-f015:**
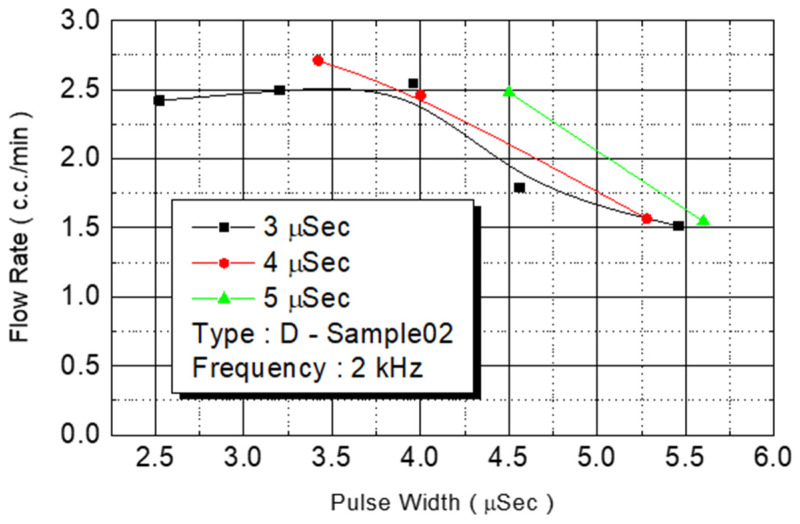
D Type-02 flow test.

**Figure 16 micromachines-12-00025-f016:**
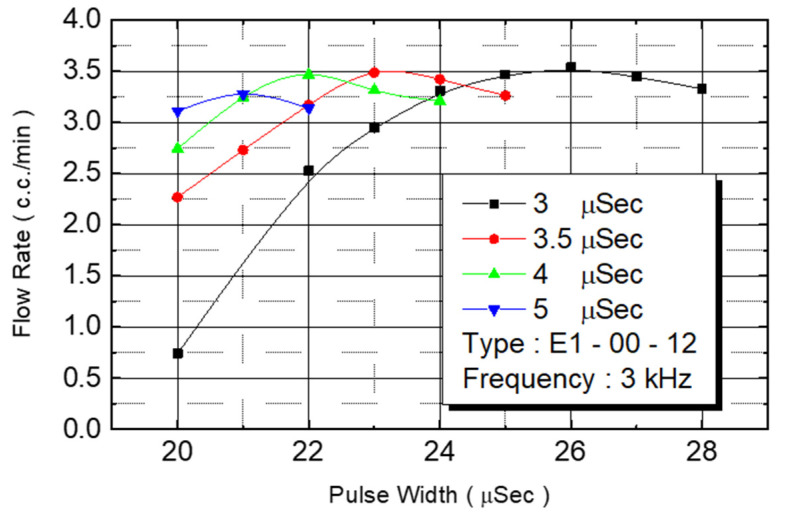
E1 type flow test.

**Figure 17 micromachines-12-00025-f017:**
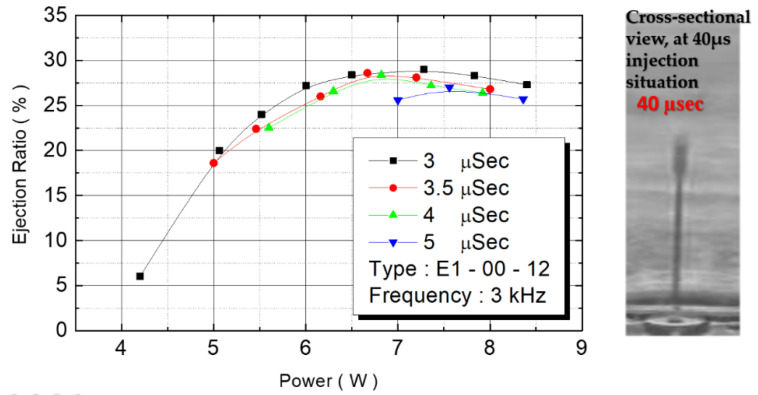
E1 type ejection rate relationship.

**Table 1 micromachines-12-00025-t001:** List of resistance measurement of single circuit resistance.

Type	A	B1	C	D	E
Heater Size (µm × µm)	105 × 105	105 × 105	78.75 × 105	105 × 78.75	132 × 132
Heater Number	100	50			
Loop Resistance (Ω)	60.1	32.1	25.6	42.2	32.3

**Table 2 micromachines-12-00025-t002:** Open pool test starting voltage results.

Voltage (V)	Pulse Width (μs)							
	2 μs	2.5 μs	3 μs	4.5 μs	5 μs	5.5 μs	6 μs	6.5 μs
A Type	-	-	-	29.56 V	28.55 V	27.73 V	26.96 V	26.23 V
B Type	21.53 V	19.64 V	18.36 V	-	-	-	-	-
C Type	21.27 V	19.50 V	18.33 V	-	-	-	-	-
D Type	18.13 V	16.53 V	15.40 V	-	-	-	-	-
E Type	27.52 V	25.38 V	23.62 V	-	-	-	-	-

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
