# Peer review of "DNA Printing Integrated Multiplexer Driver Microelectronic Mechanical System Head (IDMH) and Microfluidic Flow Estimation"

_micromachines, 2020, doi:10.3390/mi12010025_

Round 1

Reviewer 1 Report

The manuscript details the development of a MEMS head for DNA printing with the computational and experimental investigation. The manuscript can benefit from the following:

  1. English needs to be improved. 
  2. In line 97, a symbol for the sheet resistance value for the resistive layer is missing. 
  3. A description of the materials used in the experimentation through the manuscript is required. such as manufacturer, source, and model of the LED, CCD, and other chips and components.
  4. The schematic for Fig1 is unclear. A clear flow of the setup is useful for the reader to follow. 
  5. Figures with plots can be merged for better data visualization and interpretation for starting voltage (Fig's 8,10, 12) and starting energy (Fig's 9, 11, 13).
  6. The discussion lacks the summary of the experimental results and outcomes from the data to obtain optimal DNA spray based on the input parameters. 

Author Response

Dear Reviewer,

Thanks for the comments:

Comments and Suggestions for Authors

The manuscript details the development of a MEMS head for DNA printing with the computational and experimental investigation. The manuscript can benefit from the following:

1.English needs to be improved. 

Answer: Thanks for your comment, we have improved it.

2.In line 97, a symbol for the sheet resistance value for the resistive layer is missing. 

Answer: Thanks for your comment, we have modified it.

the sheet resistance of the resistance layer was 29 W / £. Therefore, the total loop resistance of Model A was the largest; it was twice that of the normal size models (Models B1, C, D, and E). The total loop resistance of Models B1, C, D, and E was approximately 29 W / £.

3.A description of the materials used in the experimentation through the manuscript is required. such as manufacturer, source, and model of the LED, CCD, and other chips and components.

Answer: Thanks for your comment, we have modified it.

For image observation, the synchronous flash method, employing light-emitting diodes (LED, Nichia NSPW500GS-K1, 3.1 V White LED 5mm), was used to generate a synchronous delayed light source. The system also used a charged-coupled device(CCD, Flir Grasshopper3 GigE GS3-PGE-50S5C-C) to capture images.

4.The schematic for Fig1 is unclear. A clear flow of the setup is useful for the reader to follow. 

Answer: Thanks for your comment, we have modified it.

5.Figures with plots can be merged for better data visualization and interpretation for starting voltage (Fig's 8,10, 12) and starting energy (Fig's 9, 11, 13).

Answer: Thanks for your comment, we have modified it.

6.The discussion lacks the summary of the experimental results and outcomes from the data to obtain optimal DNA spray based on the input parameters.

Answer: Thanks for your comment, we have modified it.

The summary of the experimental results is obtained from increasing the number of nozzles, increasing the input signal frequency, pulse width, the reduced heater size, and decreasing the input voltage (power) data to obtain the optimal DNA spray. DNA is not easy to obtain, it is estimated the amount of DNA sprayed outcomes from the data (increased number of nozzles, reduced heater size) while quickly deploying DNA according to the input parameters.

Reviewer 2 Report

The manuscript presented by Liou and coauthors introduces a 3D MEMS chip structure of DNA spray technology for placing DNA beads in the rapid array and estimating the flow rate. At current stage, the manuscript is hard to follow due to many grammatical mistakes, I recommend the authors improve the quality of the manuscript before it can be published. Also, the manuscript does not clearly manifest what problems (especially those with respect to DNA) have been addressed and why they are important.

Author Response

Dear Reviewer,

Thanks for the comments:

Comments and Suggestions for Authors

The manuscript presented by Liou and coauthors introduces a 3D MEMS chip structure of DNA spray technology for placing DNA beads in the rapid array and estimating the flow rate. At current stage, the manuscript is hard to follow due to many grammatical mistakes, I recommend the authors improve the quality of the manuscript before it can be published. Also, the manuscript does not clearly manifest what problems (especially those with respect to DNA) have been addressed and why they are important.

Answer:

(1)Thanks for the comment, we have improved it.

(2)Because DNA cannot be obtained easily, rapidly deploying DNA while estimating the total amount of DNA being sprayed is imperative. DNA printings were collected into a multiplexer driver microelectronic mechanical system head, and microflow estimation was conducted.

The summary of the experimental results is obtained from increasing the number of nozzles, increasing the input signal frequency, pulse width, the reduced heater size, and decreasing the input voltage (power) data to obtain the optimal DNA spray. DNA is not easy to obtain, it is estimated the amount of DNA sprayed outcomes from the data (increased number of nozzles, reduced heater size) while quickly deploying DNA according to the input parameters.

Round 2

Reviewer 2 Report

The authors have improved the quality of the manuscript though some sentences are still not clear.